# Drugs That Mimic Hypoxia Selectively Target EBV-Positive Gastric Cancer Cells

**DOI:** 10.3390/cancers15061846

**Published:** 2023-03-19

**Authors:** Blue-leaf A. Cordes, Andrea Bilger, Richard J. Kraus, Ella T. Ward-Shaw, Madeline R. Labott, Shinhyo Lee, Paul F. Lambert, Janet E. Mertz

**Affiliations:** McArdle Laboratory for Cancer Research, School of Medicine and Public Health, University of Wisconsin-Madison, Madison, WI 53705, USA; bcordes@wisc.edu (B.-l.A.C.); bilger@oncology.wisc.edu (A.B.); kraus@oncology.wisc.edu (R.J.K.); etward@wisc.edu (E.T.W.-S.); mlabott@wisc.edu (M.R.L.); lee827@wisc.edu (S.L.); plambert@wisc.edu (P.F.L.)

**Keywords:** deferoxamine, deferasirox, EBV, gastric cancer, lytic-induction therapy

## Abstract

**Simple Summary:**

Epstein-Barr virus (EBV) infects greater than 90% of the world’s population and remains in infected people for their lifetime, mostly in a latent form of infection. In some people, latent EBV infection leads to cancer, including EBV-associated gastric carcinomas. Lytic-induction therapy involves switching EBV from a latent to lytic form of infection so that replicating EBV can then kill EBV-infected cancer cells. Here, we examined several classes of drugs for their ability to kill EBV-positive gastric cancer cells while sparing EBV-negative ones. We found that Deferoxamine and Deferasirox, drugs approved for treating iron overload diseases, can selectively kill EBV-positive gastric cancer cells grown in culture. However, when administered to immunodeficient mice, these drugs failed to significantly induce lytic EBV infection in EBV-positive gastric tumors that had been ectopically grown in the animals.

**Abstract:**

Latent infection of Epstein-Barr virus (EBV) is associated with lymphoid and epithelial cell cancers, including 10% of gastric carcinomas. We previously reported that hypoxia inducible factor-1α (HIF-1α) induces EBV’s latent-to-lytic switch and identified several HIF-1α-stabilizing drugs that induce this viral reactivation. Here, we tested three classes of these drugs for preferential killing of the EBV-positive gastric cancer AGS-Akata cell line compared to its matched EBV-negative AGS control. We observed preferential killing with iron chelators [Deferoxamine (DFO); Deferasirox (DFX)] and a prolyl hydroxylase inhibitor (BAY 85-3934 (Molidustat)), but not with a neddylation inhibitor [MLN4924 (Pevonedistat)]. DFO and DFX also induced preferential killing of the EBV-positive gastric cancer AGS-BDneo and SNU-719 cell lines. Preferential killing was enhanced when low-dose DFX (10 μM) was combined with the antiviral prodrug ganciclovir. DFO and DFX induced lytic EBV reactivation in approximately 10% of SNU-719 and 20-30% of AGS-Akata and AGS-BDneo cells. However, neither DFO nor DFX significantly induced synthesis of lytic EBV proteins in xenografts grown in NSG mice from AGS-Akata cells above the level observed in control-treated mice. Therefore, these FDA-approved iron chelators are less effective than gemcitabine at promoting EBV reactivation in vivo despite their high specificity and efficiency in vitro.

## 1. Introduction

Epstein-Barr Virus (EBV) is a gamma herpesvirus that infects greater than 90% of the world’s population. Once EBV infects its host and undergoes lytic replication, it remains forever in the host by establishing a latent infection in a subset of memory B cells, occasionally reactivating throughout the host’s lifetime (reviewed in [1]). Latent infection is associated with cancers of lymphoid and epithelial cell origin, including most nasopharyngeal carcinomas and approximately 10% of gastric carcinomas (reviewed in [1,2]). EBV-associated gastric carcinoma (EBVaGC) is the most common of the EBV-related cancers with a worldwide incidence of 75,000–90,000 cases per year (reviewed in [3]). One hallmark of EBVaGC is DNA hypermethylation of tumor suppressors and other genes that inhibit metastasis (reviewed in [4]). Therefore, compared with EBV-negative GC, EBVaCG is associated with better prognosis due to a decreased incidence of metastasis to regional lymph nodes [5]. Furthermore, most early-stage EBVaGCs can be surgically excised without recurrence [6]. However, selectively targeting EBV-positive cancer cells for killing may serve as a potential way to treat more advanced-stage EBVaGCs and other types of EBV-associated cancers.

Lytic-induction therapy, also known as “kick and kill” and “cytolytic virus activation therapy”, involves reactivating EBV from its latent form of infection present in the tumor cells into its lytic replication cycle (reviewed in [7,8,9,10]). Reactivation begins with expression of the EBV-encoded immediate-early (IE) genes *BZLF1* (encoding Zta, also known as Z and ZEBRA) and *BRLF1* (encoding Rta, also known as R). These IE proteins then activate expression of numerous viral early genes, including *BGLF4* and *BXLF1* that encode a protein kinase (PK) and thymidine kinase (TK), respectively. The EBV lytic cycle cascade then continues with viral genome replication followed by expression of numerous late genes required for production of infectious viral particles and viremia in the patient. 

Selective killing of EBV-positive cancer cells and prevention of viremia can be achieved by addition of an antiviral nucleoside analog prodrug such as ganciclovir (GCV) or valganciclovir. To be converted to its active form, GCV requires phosphorylation by the viral PK [11], a kinase that is only expressed in cells replicating EBV or another herpesvirus. Phosphorylated GCV is then incorporated into DNA as a chain terminator, inhibiting both viral and cellular DNA replication and causing death of the reactivated EBV-positive tumor cells while sparing most EBV-negative normal tissue. Cells directly adjacent to the virally reactivated ones are also often killed by a phenomenon called the “bystander effect” [12]. Lytic reactivation efficiency does not need to be 100% to kill most or all of the cells in an EBV-positive tumor. Therefore, lytic-induction therapy may be a method to target EBV-positive tumor cells with high specificity.

Activation of *BZLF1* gene expression to induce EBV reactivation can be triggered by a variety of stimuli. These triggers include DNA-damaging agents such as gemcitabine [13,14] and histone deacetylase (HDAC) inhibitors such as valproic acid [15], romidepsin [16], and arginine butyrate (or its sodium salt) [17,18]. Arginine butyrate combined with ganciclovir has been used to treat patients with EBV-positive lymphoma refractory to other treatments with partial success in a Phase I/II clinical trial [19]. Another HDAC inhibitor, Nanatinostat, has been combined with valganciclovir in ongoing Phase Ib/II clinical trials for the treatment of relapsing and refractory EBV-positive lymphomas [20] and NPC (ClinicalTrials.gov, accessed on 5 February 2023, Identifier: NCT05166577). Likewise, gemcitabine plus valproic acid has been combined with valganciclovir to treat patients with EBV-positive NPC with partial response [21,22,23].

Hypoxia mimics that stabilize the Hypoxia Inducible Factor (HIF) family of transcription factors are another trigger for EBV reactivation [24,25,26]. Our laboratory showed drug-stabilized HIFs directly bind to a Hypoxia Response Element (HRE) present in the *BZLF1* gene promoter, Zp, thereby inducing expression of Zta [24]. We also showed that Zta protein is present in hypoxic regions of EBV-positive tumors in vivo [24]. Therefore, we proposed that drugs targeting the HIF pathway may be useful in lytic-induction therapy for treating patients with EBV-positive cancers.

Under normoxic conditions, the gene encoding HIF-1α is constitutively expressed, but the resulting protein is rapidly degraded in the cytoplasm by a process starting with hydroxylation by oxygen-dependent prolyl hydroxylase domain (PHD) proteins (reviewed in [27]). PHDs require oxygen, iron in its ferrous state (Fe^2+^), and α-ketoglutarate (KG) to perform this hydroxylation reaction on proline residues present in HIF-1α (Figure 1; reviewed in [28]). Hydroxylated HIF-1α is then recognized by the Von Hippel Lindau (VHL)-E3 ubiquitin ligase complex, leading to its ubiquitination and subsequent degradation. Under hypoxic conditions, the lack of oxygen renders the PHDs unable to hydroxylate HIF-1α, leading to its accumulation. HIF-1α then translocates to the nucleus where it binds to HREs as a heterodimer with constitutively expressed HIF-1β. In addition to inducing EBV reactivation via binding the Zp HRE in EBV-positive cells, the heterodimer also activates expression of a variety of HRE- containing cellular genes that encode proteins such as EPO, VEGF, and Glut-1 to enable adaptation to low oxygen environments (reviewed in [29]).

HIF-1α can be stabilized by a variety of drugs that target this pathway toward degradation (Figure 1). For example, Deferoxamine (DFO) and Deferasirox (DFX) are FDA-approved drugs that bind iron (reviewed in [30]), thereby chelating away one of the three co-factors required by PHDs to hydroxylate HIF-1α. BAY 85-3934 (Molidustat) inhibits the activities of the three HIF PHDs, probably via direct binding to these proteins [31]. MLN-4924 (Pevonedistat) inhibits the activity of the NEDD8-activating enzyme (NAE), thereby preventing NEDDylation of the Cullen 2 (Cul2) E3 ubiquitin ligase that complexes with VHL to ubiquitinate HIF-1α (reviewed in [32]).

Here, we screened several of these different classes of HIF-1α-stabilizing drugs for preferential killing of EBV-positive AGS-Akata gastric cancer cells relative to their matched EBV-negative AGS cells. We found that the iron chelators showed the highest specificity for selectively killing EBV-positive cells. Thus, we examined in greater detail the ability of DFO and DFX to induce EBV reactivation and cell killing both in several EBV-positive gastric cancer-derived cell lines grown in vitro and in xenografts derived from AGS-Akata cells grown in NSG mice. We describe here the results obtained from these experiments.

## 2. Materials and Methods

### 2.1. Ethics Statement

The mouse experiments were approved by the UW-Madison Institutional Animal Care and Use Committee (protocol # M005871-A06) and conducted in accordance with the NIH *Guide for the Care and Use of Laboratory Animals*. The mice were sacrificed by cervical dislocation under isoflurane anesthesia. The UW IRB classified the work with human tissues and cells as exempt.

### 2.2. Cells

AGS cells, a human gastric carcinoma cell line, were obtained from ATCC. AGS-Akata cells (also called AGS-BX1), a gift from Lindsey Hutt-Fletcher via Shannon Kenney, are AGS cells infected with the Akata strain of EBV (Akata-EBV) that contain a GFP and neomycin-resistance (NeoR) cassette inserted into the *BXLF1* gene of EBV [33]. Similarly, AGS-BDneo cells, also a gift from Lindsey Hutt-Fletcher, contain Akata-EBV with the GFP and NeoR cassette inserted into the *BDLF3* gene of EBV [34]. All three cell lines were grown in F12 medium (Thermo Fisher Scientific, Waltham, MA, USA) supplemented with 10% fetal bovine serum (FBS; Atlanta Biologicals, Flowery Branch, GA, USA) and 100 units/mL penicillin and 100 μg/mL streptomycin (pen-strep, Thermo Fisher Scientific). AGS-Akata and AGS-BDneo were additionally supplemented with 400 μg/mL of G418 to select for EBV-positive cells. SNU-719 cells (obtained from Jin-Pok Kim via Bill Sugden) are a gastric carcinoma cell line with an endogenous EBV genome [35]. These cells were maintained in RPMI-1640 medium (Thermo Fisher Scientific) supplemented with 10% FBS and pen-strep. All cell lines were authenticated by short tandem repeat analysis (WiCell, Madison, WI, USA).

### 2.3. Chemical Mimics of Hypoxia and Other Drugs

HIF-1α was stabilized with a variety of drugs: Deferoxamine (DFO, MilliporeSigma, Burlington, MA, USA); Deferasirox (DFX, ChemScene, Monmouth Junction, NJ, USA for in vitro studies, and Novartis, Basel, Switzerland for in vivo studies); BAY 85-3934 (Molidustat, Selleck Chemicals, Houston, TX, USA); and MLN-4924 (Pevonedistat, Adooq Bioscience, Irvine, CA, USA, #A11260). DFO and gemcitabine (Gem, MilliporeSigma) stock solutions were prepared in PBS for in vitro and in vivo studies. DFX, BAY-3934, and MLN-4924 stock solutions were prepared in DMSO for in vitro studies. DFX was prepared in 30% 1,2-propanediol/70% sterile 0.9% sodium chloride solution (*v*/*v*) for oral gavage treatment. Ganciclovir (MilliporeSigma) stock solutions were prepared in 0.1 M HCl.

### 2.4. Cell Viability Assays

Cells were aliquoted into 96-well plates at a density of 2000–5000 cells/well and incubated overnight at 37 °C. The medium was then removed and replaced with medium containing the indicated amount of the HIF-1α-stabilizing drug. After incubation with the drug for 24 h, the cells were washed with PBS and incubated in drug-free fresh media for another 72 h. The cells were then washed with PBS, and cell viability was assayed by measuring ATP levels using a luciferase-based CellTiter-Glo kit (Promega, Madison, WI, USA) in a VICTOR X5 2030 Multilabel Reader (PerkinElmer, Waltham, MA, USA). When cells were treated with GCV in addition to the HIF-1α-stabilizing drug, they were seeded at a density of 1000–3000 cells/well and incubated overnight at 37 °C. The cells were then incubated for 24 h with the indicated amount of DFO or DFX on days 1 and 5 of the experiment and with 20 μg/mL of GCV on the other days prior to assaying for cell viability. Luciferase activity levels for drug-treated cells were normalized to levels obtained with cells incubated in parallel in the absence of drug. *p*-values comparing cell viability of EBV-positive to EBV-negative cells were calculated with a Student’s *t*-test.

### 2.5. Western Blot Analysis

Cells derived from both culture in dishes and xenograft tumor tissue powder were lysed in SUMO buffer [36], a 1:3 mixture of Buffer 1 [5% SDS, 150 mM Tris-HCl (pH 6.8), 30% glycerol] and Buffer 2 [25 mM Tris-HCl (pH 8.3), 50 mM NaCl, 0.5% NP40, 0.5% Deoxycholate, 0.1% SDS] containing 1x protease inhibitor cocktail (Promega). Proteins from these extracts were separated by SDS-PAGE in 4%-20% polyacrylamide gels (BioRad, Hercules, CA, USA) and transferred to 0.45 μm nitrocellulose membranes (Amersham Protran^TM^, Cytiva, Marlborough, MA, USA). Membranes were blocked in 5% milk in TBST [10 mM Tris-HCl (pH 7.4), 0.15 M NaCl, and 0.1% Tween-20] (5% milk-TBST) and incubated overnight in 5% milk-TBST containing antibody specific to HIF-1α (1:1000, Abcam, Waltham, MA, USA, ab10363), Zta (BZLF1, 1:500, Santa Cruz Biotechnology, Dallas, TX, USA, sc-53904), Rta (BRLF1, 1:2500, a rabbit polyclonal antibody directed against the R peptide sequence EDPDEETSSQAVKALREMAD), EA-D (BMRF1, 1:3000, Vector Laboratories, Newark, CA, USA, #VP-E608), or VCA/p18 (BFRF3, 1:2000, East Coast Biologics, North Berwick, ME, USA, #J125). Membranes were subsequently washed in TBST, incubated in an appropriate HRP-conjugated secondary antibody, washed in TBST, incubated for 2 min in enhanced chemiluminescence (Luminata Crescendo, MilliporeSigma, #WBLUR0100), and exposed to autoradiography film (GeneMate, VWR, Radnor, PA, USA) as described previously [24]. Original blots can be found in Appendix A.

### 2.6. Immunofluorescence Assays

Cells derived from tissue culture were seeded onto glass coverslips in 10-cm dishes at a density of 1 × 10^6^ cells/dish and incubated for 24 h with or without the indicated amount of DFO or DFX. The cells were then washed in PBS, fresh media lacking drug was added to the cells, and the cells were incubated for another 24 h for AGS-Akata and AGS-BDneo cells and another 48 h for SNU-719 cells. The cells were then fixed by incubation in a 4 °C solution of acetone:methanol (1:1). After blocking for ½ h at room temperature in PBS containing 5% milk and 5% donkey serum (MilliporeSigma), the cells on the coverslips were incubated overnight at 4 °C with primary antibody to Zta (BZLF1, 1:100, Santa Cruz Biotechnology, sc-53904) diluted in the blocking solution described above. After washing in PBS, the cells were then incubated for 1 h at room temperature with a secondary antibody (donkey anti-mouse IgG with Alexa Fluor 488, 1:500, Invitrogen, Thermo Fisher Scientific, #37114). Cells were incubated in 1 μg/mL Hoechst stain (MilliporeSigma) for 15 min to detect nuclei, washed with PBS, and the coverslips were mounted onto slides using Vectashield mounting medium (Vector Labs). The slides were imaged using a Zeiss AxioImager M2 microscope with AxioImager Software version 4.8.2.

Xenograft tumors were 10 μm-sectioned onto slides, fixed in −20 °C methanol for 10 min, and blocked for 1 h in PBS containing 0.1% Tween 20 (PBS-T), 5% milk, 5% goat serum, and 5% donkey-serum. Primary antibodies to Zta (BZLF1, 1:50, Santa Cruz Biotechnology, sc-53904) and CD31/PECAM (1:100, Abcam, ab28364), diluted in blocking solution, were added to the sections and incubated overnight at 4 °C. After washing in PBS-T, the sections were incubated for 1–2 h at room temperature with the secondary antibodies [donkey anti-mouse IgG with Alexa Fluor 488, 1:300 (Invitrogen, catalog A-21202) and goat anti-rabbit IgG with Alexa Fluor 596, 1:500 (Invitrogen, catalog A-21070) for Zta and CD31, respectively]. Sections were then incubated in 1 μg/mL Hoechst stain (MilliporeSigma) for 15 min to detect nuclei, washed with PBS-T, and mounted with Vectashield mounting medium (Vector LabsNJ,). Slides were imaged as described above.

### 2.7. Generation of EBV-Positive Xenograft Tumors

Following weaning, NOD-*scid* gamma (NSG^TM^) mice were fed a 50 ppm iron (Fe) diet (Envigo, Indianapolis, IN, USA) for 4 weeks except where indicated otherwise. Mice were then injected with 1 × 10^7^ AGS-Akata cells mixed 1:1 with Matrigel (Corning, Corning, NY, USA, # 354248) into the left or both flanks as indicated. For the low iron feed study (Appendix A), blood was drawn from the maxillary vein of the mice on the indicated weeks post weaning, and hemoglobin levels were determined using a Drew Hemavet instrument. Xenograft tumors were grown for 6 weeks to a size of ~ 1–1.5 cm^3^. Mice were treated as follows: DFO (150 mg/kg by i.p. once/day for 2 days); DFX (20 mg/kg or 40 mg/kg by oral gavage once/day for 2 days); gemcitabine (60 mg/kg by i.p. once as a positive control); PBS (i.p. for the vehicle control for DFO and gemcitabine); or 30% 1,2-propanediol/70% sterile 0.9% sodium chloride solution (*v*/*v*) (by oral gavage once/day for 2 days for the vehicle control for DFX). Tumors were harvested two days after the last treatment. Approximately half of each tumor was frozen in Optimal Cutting Temperature (OCT) compound (Leica Biosytems, Deer Park, IL, USA) on dry ice and stored at −80 °C until cryosectioned for immunofluorescence staining as described above. The other half of each tumor was frozen in liquid nitrogen, crushed into tumor powder using a Cellcrusher^TM^ (Cellcrusher, Cork, Ireland), and stored at −80 °C until processed for immunoblot analysis of proteins as described above.

## 3. Results

### 3.1. EBV-positive Gastric Cancer Cells Are Preferentially Killed by Iron Chelators

*HIF1A*, the gene encoding HIF-1α protein, is expressed in all the EBV-positive gastric cancer cell lines we have examined to date [24]. While HIF-1α is degraded unless the cells are experiencing hypoxic growth conditions, drugs that target the HIF-1α-degradation pathway can enable its accumulation under normoxic conditions (Figure 1) [37,38,39]. Stabilized HIF-1α can then bind the HRE in Zp as a heterodimer with constitutively present HIF-1β, leading to induction of EBV’s lytic cycle [24]. We tested several chemical mimics of hypoxia for their ability to preferentially kill gastric cancer cells that are EBV-positive while sparing their genetically matched EBV-negative control cells (Figure 2). 

We used AGS cells which were derived from an EBV-negative gastric cancer and AGS-Akata cells that were generated by infection of AGS cells with a recombinant Akata strain of EBV containing genes encoding GFP and neomycin-resistance inserted into EBV’s non-essential early-lytic *BXFL1* gene that encodes a thymidine kinase [33]. After incubation of the cells with each of the indicated drugs for 24 h, the drug was removed, fresh medium was added, and incubation was continued for an additional 3 days before the cells were assayed for viability (Figure 2A). Under these incubation conditions, AGS cells recover from the antiproliferative effects of MLN-4924 [40], DFO, and DFX [41], allowing us to determine whether the drugs selectively target EBV-positive cells due to viral lytic reactivation. Gemcitabine, a nucleoside analog that kills cancer cells by blocking DNA replication, is a strong inducer of lytic EBV replication in both EBV-positive B cells [13] and epithelial cells [14] and served as a control.

We found that MLN-4924 (a NEDDylation inhibitor) and gemcitabine, both potent killers of cancer cells, non-selectively killed 50% of both AGS and AGS-Akata cells at concentrations of approximately 0.1 μM and 0.01 μM, respectively (Figure 2). In contrast, the PHD inhibitor BAY 85-3935 and the iron chelators DFO and DFX were less potent, but preferentially killed the EBV-positive cells. For example, after incubation with 30 μM DFO for 24 h, only 45% ± 4% of AGS-Akata cells were still viable 3 days later compared to 97% ± 4% of AGS cells (*p* = 2.5 × 10^−7^). Likewise, 24 h incubation with 30 μM DFX resulted in 37% ± 8% viability for AGS-Akata cells versus 84% ± 4% for AGS cells (*p* = 0.001). Although BAY 85-3934 also displayed preferential killing of the EBV-positive cells at 30 μM (*p* = 0.007), it was already significantly toxic to the EBV-negative AGS cells at this concentration (viability 53% ± 8%) while not very efficient at killing the EBV-positive AGS-Akata cells at a drug concentration of 10 μM (viability 83 ± 3%). Thus, we concluded that DFO and DFX were the best of these candidates for selective killing of EBV-positive gastric cancer cells. 

To further evaluate these two FDA-approved iron chelators, we examined their ability to selectively kill two additional EBV-positive cell lines (Figure 3). AGS-BDneo cells were also generated in the Hutt-Fletcher laboratory by infection of AGS cells, but with a recombinant variant of the Akata strain of EBV that contains neomycin-resistance encoding sequences inserted into the *BDLF3* gene of EBV, a non-essential late-lytic gene that encodes gp150 [34]. SNU-719 cells were directly derived from an EBV-positive gastric cancer [35]. Unfortunately, we lack a genetically matched EBV-negative cell line to which SNU-719 cells can be directly compared. In addition, SNU-719 cells exhibit a slower growth rate compared to AGS-derived cell lines [42]. Nevertheless, in comparison to AGS cells, AGS-BDneo and SNU-719 cells also exhibited preferential killing by both DFO and DFX, but to a lesser degree than AGS-Akata cells (Figure 3). For example, after incubation for 24 h with 50 μM DFX, AGS-BDneo exhibited a cell viability of 48% ± 6% (*p* = 0.03) and SNU-719 of 57% ± 6% (*p* = 0.04), compared to AGS cells with a viability of 72% ± 3%. Meanwhile, AGS-Akata cells treated likewise exhibited a viability of only 30% ± 5% (*p* = 0.001) (Figure 3B). Treatment with DFO showed similar results (Figure 3A).

Lytic-induction therapy for treating patients with EBV-associated cancers involves a two-drug regimen (reviewed in [7,9,10,43]): one drug reactivates EBV into its lytic replication cycle; and the other drug is a prodrug such as ganciclovir (GCV) that only kills cells after conversion to its active form by an EBV-encoded kinase. GCV is a nucleoside analog that is converted to its active phosphorylated state by herpesvirus-encoded kinases such as EBV’s protein kinase, but not by any known cellular kinases [11,44,45]. Thus, we next tested whether addition of GCV would enhance the preferential killing by DFX (10 μM) of our EBV-positive gastric cancer cell lines compared to AGS cells (Figure 4). The protocol used in this experiment is shown in Figure 4A. Killing by DFX was found to be statistically enhanced by co-treatment with GCV in both EBV-positive AGS-Akata and AGS-BDneo (*p* < 0.05 for both) cells while having no effect on EBV-negative AGS cells (Figure 4B). SNU-719 cells showed a similar trend, but the enhancement in killing by addition of GCV was not statistically significant (*p* = 0.09). Thus, we conclude that the enhanced killing observed by addition of GCV is due to the presence of EBV.

### 3.2. DFO and DFX Induce Synthesis of EBV Lytic Antigens in EBV-positive Gastric Cancer Cell Lines

We [24] and others [25] previously showed that incubation of EBV-positive epithelial cell lines with iron chelators leads to accumulation of HIF-1α and expression of EBV lytic antigens. However, these previous studies involved the use of very high concentrations of these iron chelators (0.2 or 1.0 mM) along with incubation with the drug for 24 h or 48 h, respectively. In contrast, we observed preferential killing of EBV-positive cells by incubating the cells for 24 h at drug concentrations as low as 10–30 μM, conditions that were minimally toxic to EBV-negative AGS cells (Figure 2 and Figure 3) and are readily achievable in patients [46].

In Figure 5, we show that expression of EBV lytic antigens is also induced at these minimally toxic dosages. For example, a 24 h incubation with 25 µM DFO was sufficient to induce high-level accumulation of EA-D (an EBV early-lytic gene product) as well as Zta in AGS-Akata and AGS-BDneo cells by 48 h (Figure 5A and 5B, respectively). Similar results were observed with 30 µM DFX (Figure 5D and 5E, respectively). SNU-719 cells required higher concentrations of the drugs, with accumulation of the EBV lytic antigens peaking later (Figure 5C,F). Also noteworthy is that a 24-h treatment with 30 µM DFX or 50 µM DFO was sufficient for high-level accumulation of p18 (an EBV late-gene product) by 48–72 h in all three EBV-positive cell lines. Thus, EBV reactivation induced by a 24-h incubation with either drug leads to a complete cycle of lytic EBV gene expression within 48–72 h despite the drug and HIF-1α no longer being present (Figure 5A–F). Therefore, we conclude that these drugs are only needed briefly at these lower concentrations to initiate EBV’s lytic replication cycle.

To determine the percentage of cells reactivated in response to low-dose DFO and DFX treatment, we performed immunofluorescence (IF) staining assays for Zta (Figure 6). A 24-h treatment with 50 µM DFO induced EBV reactivation in 33% of AGS-Akata cells. Reactivation of AGS-BDneo by DFO and both EBV-positive AGS cell lines by DFX was also efficient at ~20%-25%. However, consistent with our killing curve (Figure 3) and immunoblot (Figure 5) data, these drugs were less efficient at reactivating EBV in SNU-719 cells (8–10%). Compared to untreated cells, cell numbers were also ~3-fold (range 1.7–4.4) lower for drug-treated cells, consistent with inhibition of cell growth.

### 3.3. DFO and DFX Fail to Significantly Induce EBV Reactivation In Vivo in a Xenograft Tumor Model in Young Mice

Given these findings, we next asked whether DFO- and DFX-induced reactivation of EBV can also occur in vivo. For these experiments, we injected 1.0 × 10^7^ AGS-Akata cells into the flanks of ~6-week-old NSG mice and waited 6 weeks for the xenografts to grow to a size of 1.0–1.5 cm^3^. We then treated the mice once per day for two days with DFO (by intraperitoneal injection) or DFX (by oral gavage) and harvested the tumors two days after the last drug treatment. As a positive control, we treated mice by intraperitoneal injection with gemcitabine (GEM), an FDA-approved drug known for its potency in reactivating EBV in xenograft tumors of lymphoid [13] and gastric origin [14]. As negative controls, we treated mice with the vehicle control solutions for DFO and DFX. Figure 7 presents representative data from experiments performed to test the relative efficacies of EBV reactivation with these drugs. 

Immunoblot analysis of the harvested xenograft tumors for immediate-early (Rta) and late (p18/VCA) EBV lytic antigens revealed, as expected, that gemcitabine induced EBV reactivation efficiently, well above the background of Rta and p18 observed in tumors from mice injected with PBS as a control (Figure 7B,C). Immunofluorescence staining for Zta in flash-frozen sections of these tumors confirmed these findings (Figure 7D). CD31, a blood vessel marker, was used to show tumor architecture. Similar analyses of tumors taken from mice treated with DFO or DFX indicated the presence of Zta and Rta in many of the samples, but at significantly lower levels than observed in the gemcitabine-treated tumors (Figure 7B–D). 

In this experiment (Figure 7), mice were fed an iron-sufficient diet of 50 ppm Fe from the time of weaning. In a different experiment, we assessed the effect of dietary iron on DFO-induced lytic reactivation. At the time of tumor harvest, none of the mice fed the low-iron diets were anemic (Appendix A). Furthermore, feeding mice a low-iron diet for 10 weeks failed to enhance EBV reactivation by DFO in vivo above the level observed in mice fed regular chow (Appendix A). 

The levels of Zta and Rta found in the xenografts from the iron chelator-treated mice were not significantly higher than the background of these EBV lytic antigens observed in some of the negative control samples (Figure 7B–D). The presence of low levels of Zta and Rta in some of the negative control tumors is a reproducible finding, likely the result of EBV reactivation occurring in hypoxic regions of tumors [24]. Consistent with EBV reactivation not being significantly induced by treatment with the iron chelators, we also failed to detect the EBV late-lytic antigen p18/VCA in the DFO- and DFX-treated mice (Figure 7B,C). Thus, we conclude that treatment with DFO or DFX does not significantly increase EBV reactivation in vivo above the levels observed in control-treated animals.

## 4. Discussion

Lytic-induction therapy has potential for treating patients with EBV-associated malignancies such as EBVaGC. EBV can be induced into its lytic cycle by targeting a variety of cellular pathways, including ones that induce DNA damage, hypoxia, or alter the methylation status of EBV’s immediate-early promoters (reviewed in [7]). Here, we tested several drugs known to induce stabilization of HIF-α proteins for their ability to preferentially kill EBV-positive gastric cancer cells (Figure 2). Deferoxamine (DFO) and Deferasirox (DFX) are iron chelators FDA approved for treating patients with iron overload due to frequent blood transfusions for diseases such as thalassemia or myelodysplastic syndrome (reviewed in [47]). Incubation of 20 μM DFO or DFX for 24 h has previously been reported to induce subG1 arrest and markers of apoptosis in approximately 20% of AGS cells [41]. The NEDDylation inhibitor MLN-4924 (Pevonedistat) has been in Phase 1 clinical trials for treating patients with a variety of cancers (reviewed in [48]). It has also been shown to be cytotoxic to AGS cells [40]. The PHD inhibitor BAY 85-3934 (Molidustat) is in Phase III clinical trials for the treatment of anemia caused by chronic kidney disease [49,50]. For comparison, we also tested the chemotherapeutic drug gemcitabine that is used to treat patients with a variety of cancers (reviewed in [51]). 

We found that Gemcitabine and MLN-4924 efficiently killed our cancer cell lines regardless of the presence of EBV. In contrast, all three of the drugs that targeted the activity of the HIF-1α PHDs either directly (BAY 85-3934) or indirectly (DFO, DFX) preferentially killed EBV-positive cells. However, BAY 85-3934 displayed a very narrow therapeutic index with only 20% preferential killing at 10 μM concentration, yet toxic to both EBV-positive and -negative cells at 30 μM. Therefore, we focused our attention on the FDA-approved iron chelators, DFO and DFX, which were effective at preferentially killing EBV-positive cells at doses readily achievable in patients [46]. Importantly, we were able to identify doses of DFO and DFX that significantly killed EBV-positive cells while inducing minimal-to-no killing of the matched EBV-negative cells (Figure 3). DFO is known to cause mitochondrial dysfunction and decreased ATP production. However with removal of DFO, mitochondrial function returns to baseline within 72 h, indicating that DFO’s effects are reversible [52]. Given we assayed for cell viability at 96 h (72 h after removal of DFO), DFO should no longer have been affecting ATP production. Thus, the decreased ATP measured in the EBV-positive lysates (Figure 3) was likely due to the decreased number of viable cells secondary to EBV lytic reactivation. In support of this conclusion was our finding that addition of GCV further enhanced preferential killing of EBV-positive cells (Figure 4).

We also found that incubation with low dosages of these iron chelators for 24 h was sufficient to induce the complete lytic EBV replication cycle as indicated by synthesis of the late lytic protein p18/VCA within 48 h (Figure 5). For example, we observed p18 protein at 48 h in AGS-Akata cells treated with doses as low as 25 μM for DFO and 10 μM for DFX. Synthesis of the late-lytic protein p18/VCA indicates that at least some of the EBV-positive cells in Figure 3 are dying due to lytic reactivation of EBV. In contrast, Yiu et al. failed to observe p18 in AGS-Akata cells incubated for 48 h with DFX and only tiny amounts of this late-lytic protein at concentrations of DFO of 250 μM and above [25]. A likely explanation for this difference between our results is that we withdrew the iron chelators from the medium after 24 h, while they retained them throughout the duration of their experiments. Their prolonged incubation with iron chelators likely led to inhibition of viral and cellular DNA synthesis and, thus, lack of EBV late-gene expression and an increase in non-specific cell death, respectively. Consistent with this hypothesis, they observed 30–40% death of EBV-negative AGS cells with concentrations of iron chelator as low as 1.25 µM at 48 h. These findings indicate the importance of keeping the treatment with iron chelator brief, i.e., long enough to initiate EBV reactivation, but short enough to attenuate non-specific cell toxicities. Fortunately, this approach should be feasible given Yiu et al. also showed that incubation of the EBV-positive NPC HA cell line with their novel iron chelator C7 for as little as 2–4 h is sufficient to induce synthesis of Zta protein in vitro [53]. Possibly, the use of brief incubation times may enhance the selectivity for preferential killing of EBV-positive cells by HIF-1α-stabilizing drugs, including by BAY 85-3934 which is not dependent upon the animal being anemic to inhibit the activity of PHDs. 

Treatment with iron chelators for brief incubation times may also affect the activity of enzymes in the superfamily of Fe(II) α-ketoglutarate-dependent dioxygenases, where iron is a cofactor. This superfamily not only includes PHDs, but also the Ten-Eleven Translocation (TETs) DNA demethylases and Jumoji (Jmj) Domain-containing Histone-Lysine Demethylases (KDMs) (reviewed in [54]). The activity of these demethylases are relevant, because EBVaGC is associated with DNA hypermethylation of both genomic and viral DNA, the latter of which maintains latent infection and is required for BZLF1-induced lytic reactivation (reviewed in [55]). Chelation of iron by DFO inhibits the activity of TETs and Jmj-KDMs in HEK 293 cells, resulting in increased methylated DNA and histone methylation, respectively. However, replacement of iron restores both the enzyme activities of TETs and Jmj-KDMs and DNA and histone methylation levels to that observed in untreated cells within 4 hours [56]. Thus, our protocol of removing the DFO after 24 h and replacing with iron-replete media likely alleviated the effect that DNA and histone methylation status may have played in EBV lytic reactivation. In addition, EBV infection of gastric cancer cells MKN7 has been shown to downregulate TETs at the transcriptional level, especially TET2 [57]. If this were also true in the GC cell lines studied here, treatment with iron chelators would have minimally affected the activities of these already down-regulated TETs. Alternatively, if the EBVaGC cell lines retained significant TET activity, as has shown to be the case for EBV-infected telomerase-immortalized normal oral keratinocytes (NOK-Akata) [58], inhibition of TET activity by iron chelators may potentiate lytic reactivation by BZLF1. Wille et al. showed that knockdown of endogenous TET activity promotes BZLF1 lytic reactivation in NOK-Akata cells [58].

For lytic-induction therapy to be effective, a significant percentage of the cells (albeit considerably less than 100% because of bystander killing [12]) need to be reactivated so synthesis of the EBV-encoded protein kinase required to convert the prodrug GCV or valganciclovir to its active form occurs. By incubating with 1000 μM DFO for 48 h, Yiu et al. were able to achieve 64% reactivation of AGS-BDneo cells [25]. We showed 33% and 21% reactivation of AGS-Akata and AGS-BDneo cells, respectively, by incubation for 24 h with 50 μM DFO (Figure 6), a 20-fold lower dose. Treatment of SNU-719 cells with 50 µM DFO or 30 µM DFX resulted in 8%-10% reactivation, similar to what has been observed with gemcitabine treatment of SNU-719 cells [14]. We further showed that addition of GCV enhanced preferential killing of EBV-positive cells by DFX (Figure 4). Although the enhancement of killing by addition of GCV was not statistically significant in SNU-719 cells in our study, Lee et al. previously reported that GCV combined with Gemcitabine did enhance killing of SNU-719 cells using a five-fold higher dosage of GCV than we used here [14]. Given that gemcitabine has shown preliminary promise as an agent for lytic-induction therapy [13,14], these findings suggested the FDA-approved iron chelators DFO and DFX may have potential in vivo as well.

We therefore examined the efficacy of DFO and DFX to induce synthesis of EBV lytic antigens in AGS-Akata xenografts grown in NSG mice (Figure 7). We found that neither DFO nor DFX significantly induced synthesis of the immediate-early proteins Zta and Rta above the background levels observed in tumors from control-treated mice. Lowering the amount of iron in the diet for 10 weeks immediately following weaning did not lower the level of mouse hemoglobin and failed to significantly enhance EBV reactivation in vivo by DFO (Appendix A). The mouse diet also included ascorbate, a vitamin lacking in our cell media that reduces iron to its ferrous state so it can be utilized by PHDs and TETs. Thus, the presence of ascorbate in the mice may have further attenuated DFO- and DFX-induced EBV reactivation in vivo. Given these negative findings, we did not co-treat with iron chelators and GCV. In contrast, gemcitabine consistently induced greatly enhanced in vivo synthesis of Rta and even induced synthesis of the late-lytic protein p18/VCA (Figure 7). This finding for gemcitabine treatment of AGS-Akata xenografts is similar to a previously reported finding performed with SNU-719 xenografts [14].

The failure of DFX and DFO to induce significant levels of EBV reactivation in the AGS-Akata xenografts grown from cells injected ectopically into the flanks of mice is likely to be due to insufficient iron reduction in the target cells. DFO and DFX were administered systemically via intraperitoneal and oral delivery, respectively. Iron stores in mice include circulating and tissue ferritin and heme iron in erythrocytes. The 2-day systemic treatments with DFO or DFX were likely not adequate to lower the concentration of iron within the tumor cells to the level necessary to stabilize HIF-1α for induction of Zta synthesis.

EBV reactivation might be successfully induced with DFO or DFX, if these iron chelators could be more directly targeted to the tumor. In patients, EBV-positive gastric carcinoma generally occurs in the proximal part of the stomach (reviewed in [59]), which is accessible by endoscopy. Inoperable gastric tumors have been successfully treated by direct injection of drugs such as carbon-adsorbed methotrexate [60] or cisplatin/epinepherine gel [61] in these case studies. Similarly, direct injection of tumors could be done with DFO or DFX to overcome the problem of high systemic levels of iron in the animals, but we lack a mouse model for *in situ* EBVaGC. Alternatively, it may be possible to target DFO or DFX to tumors by drug delivery systems. For example, when DFO and the HIF-1α inhibitor YC1 were formulated into transferrin-decorated liposome nanoparticles and targeted to transferrin receptor-expressing pancreatic cancer xenografts in mice, xenograft tumor regression was observed [62]. Therefore, intratumoral injection or liposome nanoparticle-based targeting of these iron-chelation drugs warrants further investigation as a potential treatment for EBV-positive gastric cancers and NPCs. In addition, other drugs such as direct PHD inhibitors that stabilize HIF-αs regardless of iron level may be useful for lytic-induction therapy either alone or in combination with other lytic-inducing agents such as HDAC inhibitors.

## 5. Conclusions

Low doses of DFO and DFX preferentially kill EBV-positive gastric cancer cells compared to EBV-negative gastric cancer cells in vitro. These doses of DFO and DFX also induce lytic reactivation in EBV-positive gastric cell lines in vitro. However, in contrast to gemcitabine, neither DFO nor DFX significantly induced synthesis of lytic EBV proteins in AGS-Akata xenografts above the level observed in control-treated mice. Therefore, these FDA-approved iron chelators are considerably less effective than gemcitabine at promoting EBV reactivation in vivo despite their high specificity and efficiency in vitro. Possibly, these drugs may work in patients that happen to be iron-deficient or in cases where the drug can be targeted to the tumor.

## Figures and Tables

**Figure 1 cancers-15-01846-f001:**
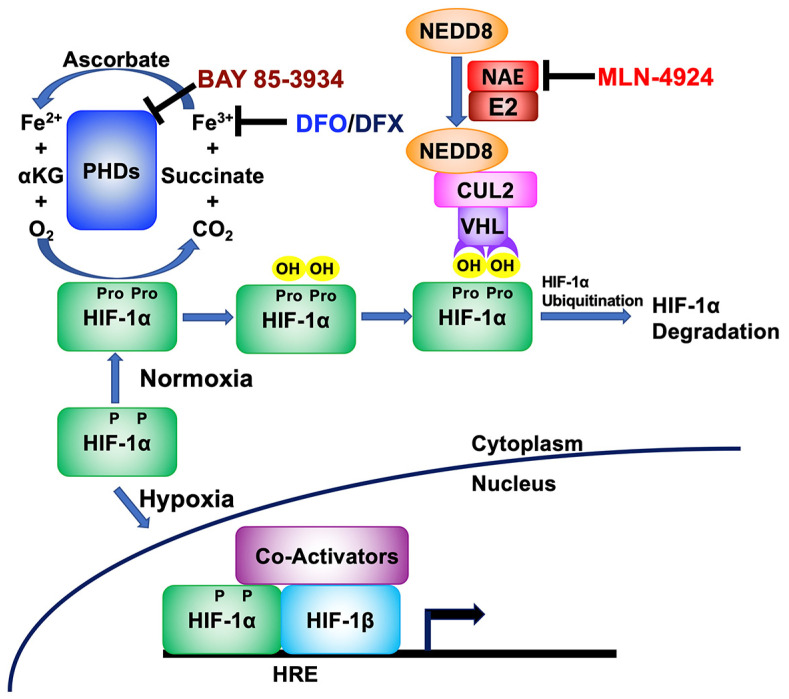
Schematic showing where some drugs stabilize HIF-1α by targeting different parts of the hypoxia signaling pathway. Under normoxic growth conditions, HIF-1α protein is synthesized. However, it is then hydroxylated on proline residues (Pro) by prolyl-hydroxylases (PHDs), cellular enzymes that use α-ketoglutarate (αKG), ferrous iron (Fe^2+^), and oxygen as co-factors. The hydroxylated form of HIF-1α is then recognized by the VHL/Cullen 2 (CUL2) complex for ubiquitination, if Cullen 2 has been NEDDylated by the NEDD8-activating enzyme (NAE). Finally, ubiquitinated HIF-1α is then rapidly degraded via the cellular proteasome complex. However, if this pathway is disrupted, HIF-1α protein accumulates, is translocated to the nucleus, dimerizes with constitutively stable HIF-1β, and binds hypoxia-response elements (HREs) located within cellular and EBV genomes. DFO and DFX chelate ferric iron (Fe^3+^), preventing its conversion to Fe^2+^; BAY 85-3934 directly binds PHDs, inhibiting their activity; and MLN-4924 directly binds NAE, inhibiting NEDDylation of Cullen 2.

**Figure 2 cancers-15-01846-f002:**
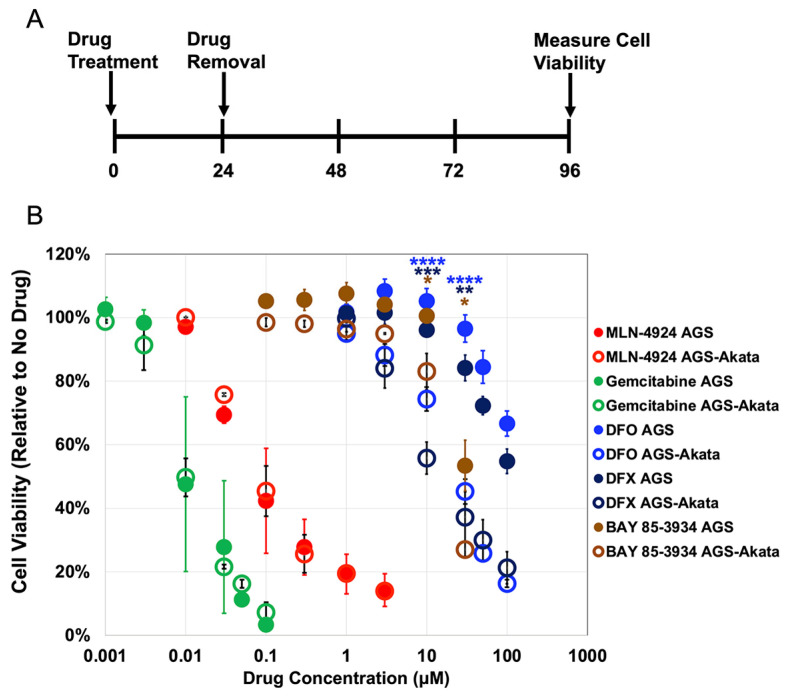
Cell viability of EBV(+) AGS-Akata versus EBV(-) AGS cells following incubation with a variety of hypoxia mimics versus gemcitabine. (**A**) Protocol used for treating cells for 24 h with the indicated agents followed by assaying for cell viability at 96 h. (**B**) Effects of various dosages of gemcitabine (green), MLN-4924 (red), DFO (blue), DFX (navy), and BAY 85-3934 (brown) on cell viability of AGS (closed circles) and AGS-Akata (open circles). Error bars indicate ± S.E.M.; *n* = 3–9; *p*-values for EBV-positive cells compared to AGS cells: *, *p* ≤ 0.05; **, *p* ≤ 0.005; ***, *p* ≤ 5 × 10^−4^; ****, *p* ≤ 5 × 10^−5^.

**Figure 3 cancers-15-01846-f003:**
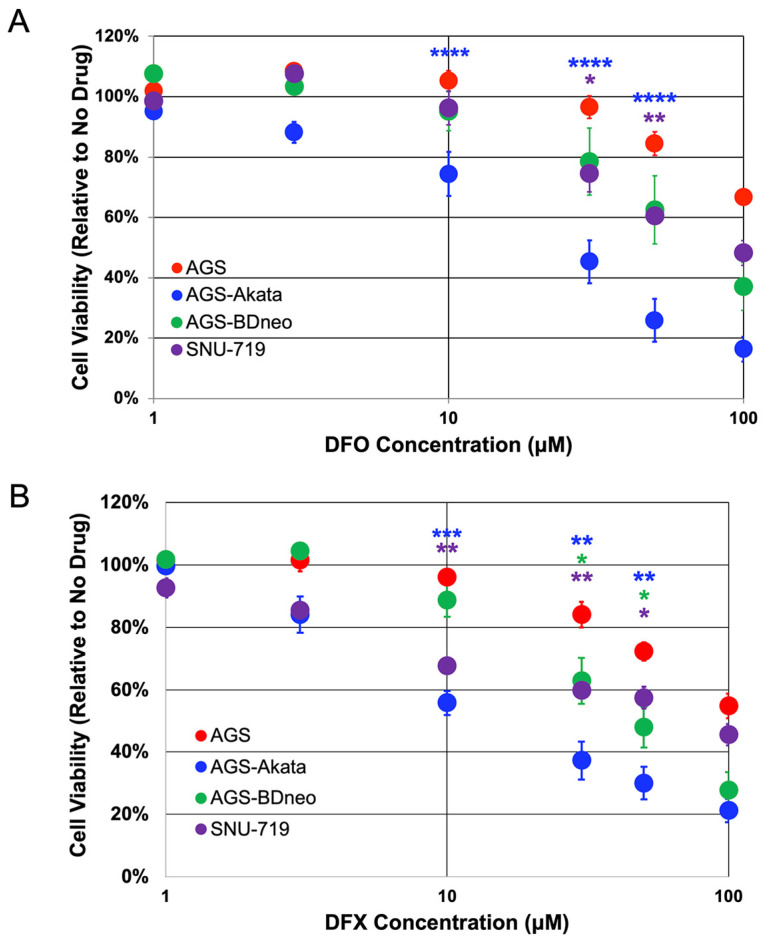
Cell viability of EBV-positive gastric cancer cell lines treated with DFO and DFX. EBV-positive cell lines AGS-Akata (blue), AGS-BDneo (green), and SNU-719 (purple) and EBV-negative AGS cells (red) were treated with the indicated concentrations of DFO (**A**) or DFX (**B**) for 24 h and then without drug for an additional 72 h. *n* = 4–9; *, *p* ≤ 0.05; **, *p* ≤ 0.005; ***, *p* ≤ 5 × 10^−4^; ****, *p* ≤ 5 × 10^−5^ for EBV^+^ cells compared to EBV^-^ AGS cells.

**Figure 4 cancers-15-01846-f004:**
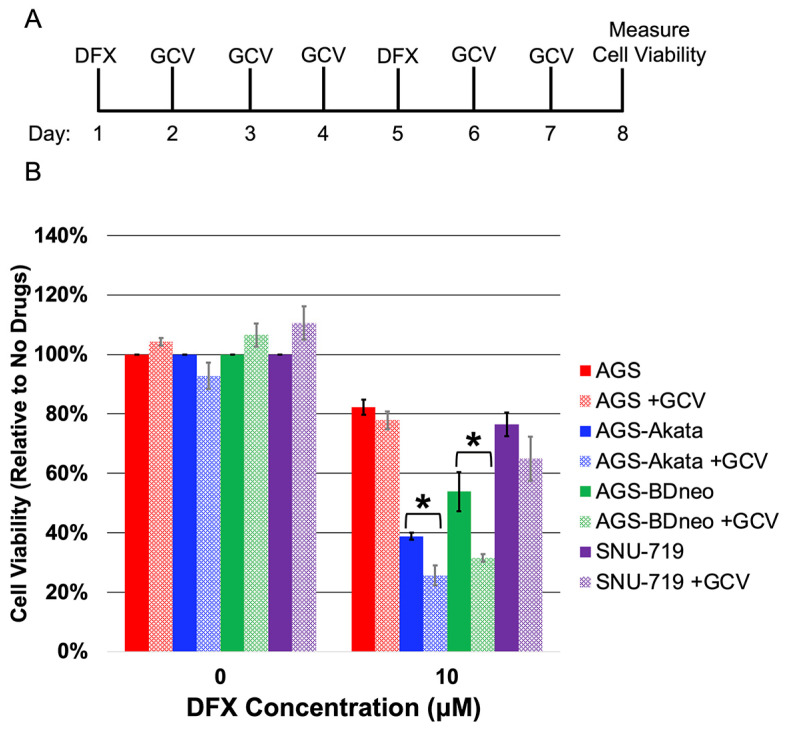
Ganciclovir enhances preferential killing of EBV-positive gastric cancer cells by DFX. (**A**) Schematic showing protocol used to examine effects of addition of GCV on killing by DFX. Cells were incubated for 24 h with 10 µM DFX (or DMSO as the diluent control) on the 1st and 5th days of the experiment and 20 μg/mL GCV (or the HCl diluent control) on each of the other days. Cell viability was determined on day 8. (**B**) Cell viability of EBV-negative AGS (red) cells and EBV-positive AGS-Akata (blue), AGS-BDneo (green), and SNU-719 (purple) after incubation with or without the indicated drugs as shown in panel A. Data for each cell line were normalized to the percentage cell viability observed when these cells were incubated in parallel with neither drug. Error bars represent ± S.E.M.; *n* = 3; *, *p* ≤ 0.05.

**Figure 5 cancers-15-01846-f005:**
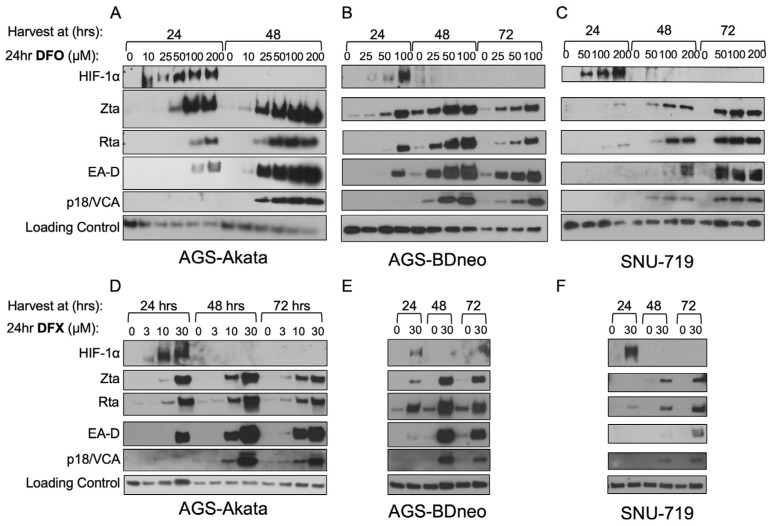
Low-dose DFO and DFX induce synthesis of EBV early- and late-lytic proteins as well as immediate-early proteins in EBV(+) gastric cancer cell lines. AGS-Akata (**A**,**D**), AGS-BDneo (**B**,**E**), and SNU-719 (**C**,**F**) cells were incubated for 24 h with the indicated concentrations of DFO (**A**–**C**) or DFX (**D**–**F**). After the drug was removed, the cells were incubated for an additional 0, 24, or 48 h prior to harvesting. Accumulation of the EBV immediate-early (Zta, Rta), early (EA-D/BMRF1), and late (p18/VCA) lytic viral antigens was determined by immunoblot analysis. (*n* = 2, representative blots shown). Treatment of AGS-Akata cells at concentrations of 0 μM and 200 μM (Panel A) was published previously in Kraus et al. 2017 [24].

**Figure 6 cancers-15-01846-f006:**
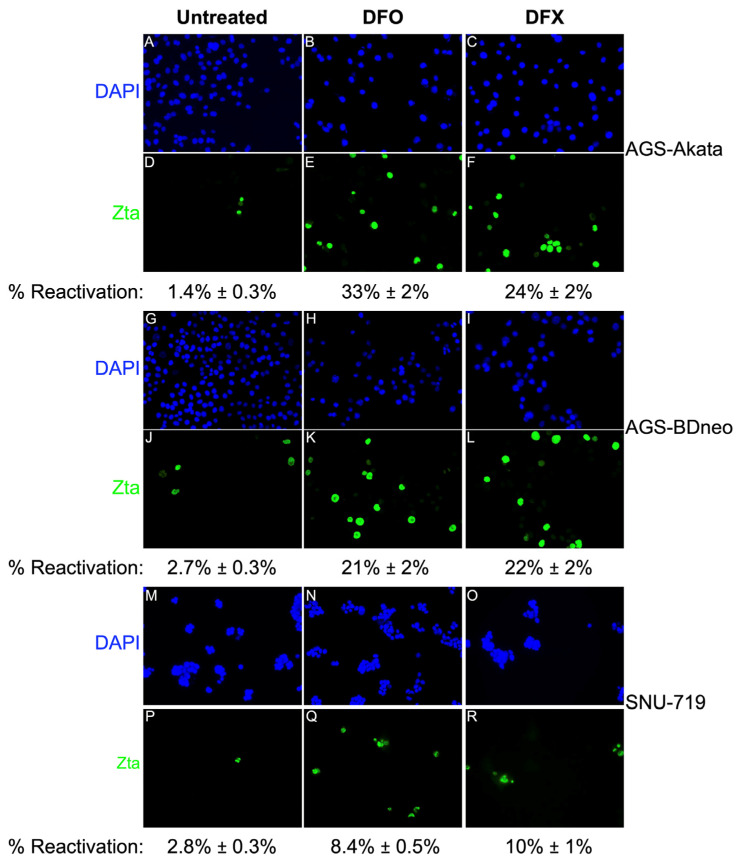
Efficiencies of viral reactivation following 24 h incubation of EBV(+) gastric cancer cells with a low dose of DFO or DFX. The cells were incubated for 24 h with 50 μΜ DFO (**B**,**E**,**H**,**K**,**N**,**Q**) or 30 µM DFX (**C**,**F**,**I**,**L**,**O**,**R**). After removal of the drug, incubation was continued for another 24 h [AGS-Akata (**A**–**F**); AGS-BDneo (**G**–**L**)] or 48 h [SNU-719, (**M**–**R**)] prior to fixation and staining with DAPI (for nuclear DNA) and a Zta-specific antibody. Five-to-ten fields of cells were examined in each dish from 2–3 separate experiments to determine the mean percentages of Zta+ cells shown in representative images photographed at 20X magnification. Error bars represent ± S.E.M.; *p* ≤ 5 × 10^−5^ for treatment compared to control.

**Figure 7 cancers-15-01846-f007:**
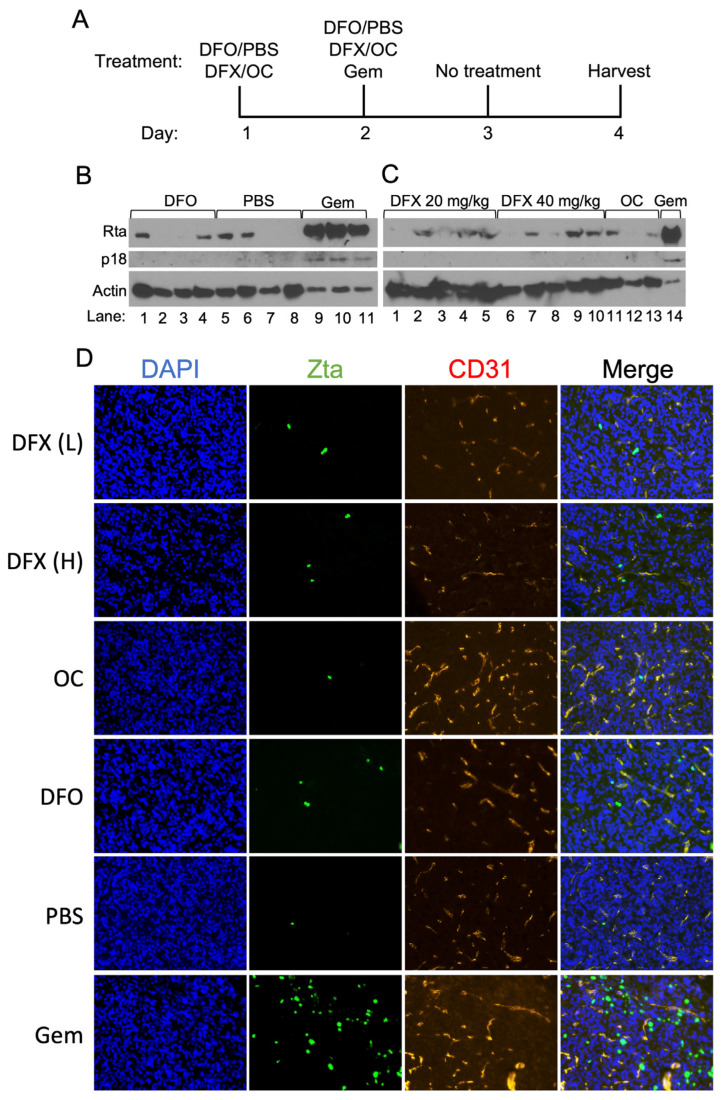
Effects of DFO and DFX treatment of AGS-Akata xenografts grown in immuno-suppressed mice. 1 × 10^7^ AGS-Akata cells were injected into the left flank of NSG mice. When the xenografts reached ~150 mm^3^ in size, the mice were treated as described in (**A**) with DFO, a low (L, 20 mg/kg) or high (H, 40 mg/kg) dose of DFX, gemcitabine, PBS, or the DFX vehicle (OC). The tumors were harvested two days after the last treatment. (**B**,**C**) Half of each tumor was homogenized and processed by immunoblot analysis for Rta, p18/VCA, and actin as a loading control. For gemcitabine-treated tumor samples, 8-fold less extract was loaded into the gel. (**D**) The other half of each tumor was flash-frozen, 10 μm sections taken, and stained for Zta (green), the blood vessel marker CD31 (red), and DAPI (blue) to detect nuclear DNA. Shown here are representative data photographed at 20X magnification from 3 separate experiments performed with DFO and 1 with DFX.

## Data Availability

The data presented in this study are available in the article.

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
