# Peer review of "Drugs That Mimic Hypoxia Selectively Target EBV-Positive Gastric Cancer Cells"

_cancers, 2023, doi:10.3390/cancers15061846_

Round 1
Reviewer 1 Report
I thank the authors for the opportunity to review their manuscript entitled: "Drugs That Mimic Hypoxia Selectively Target EBV-Positive Gastric Cancer Cells".
The topic is certainly of interest to the readership and this study builds upon the authors' previously published work.
Comments (by order of appearance):
(1) Introduction - Please consider adding some more background on EBVaGC (mainly on oncogenesis, prognosis and therapeutic implications).
Methods:
(2.2) Have the cells (purchased and donated) been validated (e.g STR)?
(2.4)
(A) DFO has been show before (in similar doses, in other cells lines) to result in growth retardation, mitochondrial inhibition and decreased ATP production. Have you assessed cell viability in other, non ATP-dependent methods as well?
(B) It is important to note even if the decreased ATP readout from the CellTiterGlo corresponds with cell count - it does not necessarily suggest cytotoxicity, as it can also correspond with growth delay.
(2.7) Can you clarify the treatment schedule? If I read this correctly - the tumors were treated for 2 days and harvested 2 days after that. is that correct? Maybe include a schematic as in figures 2A, 4A.
(3.1)
(A) Have you done any analysis to indicate whether this is indeed cell death vs. growth delay? (for example, CellTox, Annexin/PI cytometry, Tryphan Blue, WB for Cleaved caspase, etc.)
(B) Please include calculated IC50s with 95%CI (which are better to use when comparing dose-response curves and looking for significant differences)
(C) Are the growth rates of all 4 cell lines the same? if not, perhaps GR50 comparisons are more appropriate.
(D) in Figure 4B - in order to understand the impact of adding DFO to GCV treatment, please include GCV dose-response curves. Ideally a combenefit analysis should be considered in order to determine if the combination is synergistic, additive or antagonistic.
(3.2)
Figure 5 -
(A) Were the antigens assessed from the media or from cells?
(B) Please include an AGS cell line (negative control)
(C) Loading controls are not equal
Figure 6 -
(A) Since AGS-Akata and AGS-BDneo are akso expressing GFP - can this be affecting your measurements?
(B) Have you tried measuring GFP as a way to assess viral expression
(C) Have you done any qPCR to assess viral load?
(3.3)
What was the rationale for choosing AGS-Akata for the in-vivo experiment over the endogenously EBV expressing SNU-719?
Have you considered assessing this in a syngeneic model? ( which also allow investigating the immune-response aspects of hypoxia mimetics in this setting.)
Author Response
Review 1:
Thank you for taking the time to carefully read over our manuscript and to provide a detailed list of suggestions for improvements to it. Indicated below are our responses to each comment.
(1) Introduction - Please consider adding some more background on EBVaGC (mainly on oncogenesis, prognosis and therapeutic implications).
We have now added two sentences with supporting references to include more of this background information (lines 45 - 49).
Methods:
(2.2) Have the cells (purchased and donated) been validated (e.g STR)?
We now indicate in the Methods that the cell lines have been validated by short tandem repeat (STR) analysis (lines 157 - 158).
(2.4) (A) DFO has been shown before (in similar doses, in other cells lines) to result in growth retardation, mitochondrial inhibition and decreased ATP production. Have you assessed cell viability in other, non-ATP-dependent methods as well? (B) It is important to note even if the decreased ATP readout from the CellTiterGlo corresponds with cell count - it does not necessarily suggest cytotoxicity, as it can also correspond with growth delay.
We agree that ATP-based cell titer assays do not distinguish between cytotoxicity and growth delay. However, previous studies have already documented the cytotoxic effects of MLN-4924 (Lan et al. Nature 2016), DFO, and DFX (Kim et al Tumor Biology 2016) on AGS cells using non-ATP-dependent methods. We now cited these studies (lines 268-271 and 446-450). We also agree that DFO is known to cause mitochondrial dysfunction and decreased ATP production. However, with removal of DFO, mitochondrial function returns to baseline within 72 hours, indicating that DFO’s effects on the cell are reversible (Rensvold et al Cell Reports 2013). Since we are assaying for viability at 96 hours (72 hours after removal of DFO), DFO is no longer affecting ATP production, and the decreased ATP being measured in our studies is, thus, probably due to decreased number of viable cells. We have now added this point to the Discussion (lines 463-485). Given we also see synthesis of the late-lytic protein p18/VCA (Fig. 5), we know that at least some of the EBV-positive cells are dying due to lytic reactivation of EBV. This point has also now been added to Discussion (lines 490-491). Though some of the reduction in ATP readout could be due to non-EBV-related effects of the drugs on rate of growth of the cells and non-specific cytotoxicity, these effects are corrected for by comparing the effects of the drugs on EBV-positive cells versus their matched EBV-negative AGS cells.
(2.7) Can you clarify the treatment schedule? If I read this correctly - the tumors were treated for 2 days and harvested 2 days after that. is that correct? Maybe include a schematic as in figures 2A, 4A.
We agree and have now added a schematic to indicate more clearly the treatment schedule (new Fig. 7A).
(3.1) (A) Have you done any analysis to indicate whether this is indeed cell death vs. growth delay? (for example, CellTox, Annexin/PI cytometry, Tryphan Blue, WB for Cleaved caspase, etc.) -
Please see above response under section 2.4. In Figure 2, we were simply screening for drugs that would preferentially kill (and/or inhibit growth of) EBV-positive gastric cancer cells compared to EBV-negative AGS cells regardless of the mechanism. By removing the drug after 24 hours and assaying at 96 hours, we hoped to capture cell death specifically due to reactivation of EBV. We later show in Figure 5 this is probably the case because we observed at 48 hours expression of the EBV late-lytic protein VCA/p18 indicative of EBV-induced cytotoxicity. Furthermore, we observe fewer cells at 48-72 hours in the drug-treated cells in Figure 6. We have now quantified these data, adding a sentence to the text (lines 374 - 375). If the in vivo mouse experiments had given encouraging results, we would have been encouraged to devote some time to analyzing in detail the mechanism(s) for the lower number of viable cells present in the drug-treated EBV-positive cells.
(B) Please include calculated IC50s with 95%CI (which are better to use when comparing dose-response curves and looking for significant differences)
We are unable to calculate accurate IC50s because we didn’t test the higher doses of the drugs of the iron chelators and Molidustat required to kill 100% of the cells. Therefore, we are missing the right sides of the curves approaching zero and then plateauing there. We also lack enough data points around 50% viability to allow accurate calculation of IC50s.
(C) Are the growth rates of all 4 cell lines the same? if not, perhaps GR50 comparisons are more appropriate. –
We and others (e.g., Min et al International Journal of Biological Sciences 2019) observe that SNU719 cells grow at a slower rate than do the AGS-derived cell lines . This may well be part of the reason they reactivated more slowly following drug treatment (Fig. 5). We have now added a sentence (lines 295-296) to indicate this point.
(D) in Figure 4B - in order to understand the impact of adding DFO to GCV treatment, please include GCV dose-response curves. Ideally a combinate analysis should be considered in order to determine if the combination is synergistic, additive or antagonistic.
We chose in this experiment to use a dosage of GCV in the mid-range of dosages determined by others in the EBV field to be appropriate in combination with other drugs that reactivate EBV. Given GCV doesn’t affect cells unless the EBV protein kinase protein is present, one wouldn’t expect to observe synergistic, additive, or antagonistic effects.
(3.2)
Figure 5 - (A) Were the antigens assessed from the media or from cells?
In section 2.5, we indicate clearly that we assay cells, not media, for EBV protein levels.
(B) Please include an AGS cell line (negative control)
We previously reported the effects of DFO in AGS cells (Kraus et al., 2020). Given AGS cells don’t synthesize any EBV proteins, we see little need to repeat this negative control here.
(C) Loading controls are not equal
We agree, but that result is expected, in part, given there are fewer cells and, thus, less protein, in many of the drug-treated samples. We attempted to load similar amounts of protein per well, but often could not do so because of a lower concentration of protein in the sample. Where difference in the loading controls exist, the reader can take those differences into consideration as they examine the data.
Figure 6 - (A) Since AGS-Akata and AGS-BDneo are also expressing GFP - can this be affecting your measurements? –
GFP is not observed in our immunofluorescence assays since the cells have been fixed and permeabilized during processing (e.g., see untreated AGS-Akata cell control). Also, AGS-BDneo cells do not contain GFP.
(B) Have you tried measuring GFP as a way to assess viral expression. (C) Have you done any qPCR to assess viral load?
We viewed our measurements of EBV IE, early, and late-lytic proteins as providing more informative assays to understand what was happening over time in the drug-treated cells.
(3.3) What was the rationale for choosing AGS-Akata for the in-vivo experiment over the endogenously EBV expressing SNU-719?
We choose AGS-Akata because AGS could have been used later as a matched EBV-negative control to examine the effects of our drugs, together or in the absence of GCV, on the rate of growth of the xenografts in the NSG mice. However, given the failure of DFO and DFX to significantly induce EBV reactivation above the level observed in the control-treated mice, it didn’t make sense to proceed with this expensive, planned subsequent experiment. Unfortunately, to the best of our knowledge, an EBV-negative derivative of SNU-719 cells does not exist to provide a properly matched control for xenograft experiments performed with SNU-719 cells. It would be great if someone attempted to cure SNU-719 cells of EBV; however, it may or may not be possible given EBV-infected cells are sometimes dependent upon the expression of some EBV genes or miRNAs for their survival.
Have you considered assessing this in a syngeneic model? (which also allow investigating the immune-response aspects of hypoxia mimetics in this setting.)
To the best of our knowledge, there is no syngeneic model for studying EBVaGC in mice. It would be terrific if such a model existed.
Given our responses above, we hope you now find our revised manuscript acceptable for publication in Cancers.

Reviewer 2 Report
The authors present an approach to study the antiviral effect of some drugs (iron chelators) that may be capable to interfere in different parts of the hypoxia signaling pathway and stabilize HIF-a. The work is interesting and well presented. I have a minor comment that the figures' resolution and size should be greatly improved. Apart of that the manuscript can be accepted.
Author Response
Reviewer 2
Open Review
Comments and Suggestions for Authors:
The authors present an approach to study the antiviral effect of some drugs (iron chelators) that may be capable to interfere in different parts of the hypoxia signaling pathway and stabilize HIF-a. The work is interesting and well presented. I have a minor comment that the figures' resolution and size should be greatly improved. Apart of that the manuscript can be accepted.
Thank you for your review. We have now both enlarged and used higher resolution for all the figures in the manuscript.
Reviewer 3 Report
The presented topic is interesting and and suitable for cancer. Nevertheless, in my opinion authors they, conceive the problem too narrowly. Most probably, Deferoxamine and Deferasirox can have significantly wider range of effect relevant to gene expression. Jumonji domain histone demethylases and TET proteins are α-ketoglutarate and Fe(II)-dependent dioxygenases. Decrease Fe(II) level by chelator can increase DNA and histone methylation. In the discussed hypoxia condition HIF-1a and TET proteins collaborate gene expression.
Badal, S.; Her, Y.F.; Maher, L.J., 3rd. Nonantibiotic Effects of Fluoroquinolones in Mammalian Cells. J Biol Chem 2015, 290, 22287-22297, doi:10.1074/jbc.M115.671222.
Palei, S.; Weisner, J.; Vogt, M.; Gontla, R.; Buchmuller, B.; Ehrt, C.; Grabe, T.; Kleinbölting, S.; Müller, M.; Clever, G.H.; et al. A high-throughput effector screen identifies a novel small molecule scaffold for inhibition of ten-eleven translocation dioxygenase 2. RSC Med Chem 2022, 13, 1540-1548, doi:10.1039/d2md00186a.
Chen, H.F.; Wu, K.J. Epigenetics, TET proteins, and hypoxia in epithelial-mesenchymal transition and tumorigenesis. Biomedicine (Taipei) 2016, 6, 1, doi:10.7603/s40681-016-0001-9.
In presence of ascorbate, iron deficiency can not be limited factor for the activity of TET proteins and probably for other α-ketoglutarate and Fe(II)-dependent dioxygenases.
Dickson, K.M.; Gustafson, C.B.; Young, J.I.; Züchner, S.; Wang, G. Ascorbate-induced generation of 5-hydroxymethylcytosine is unaffected by varying levels of iron and 2-oxoglutarate. Biochem Biophys Res Commun 2013, 439, 522-527, doi:10.1016/j.bbrc.2013.09.010.
Yin, R.; Mao, S.Q.; Zhao, B.; Chong, Z.; Yang, Y.; Zhao, C.; Zhang, D.; Huang, H.; Gao, J.; Li, Z.; et al. Ascorbic acid enhances Tet-mediated 5-methylcytosine oxidation and promotes DNA demethylation in mammals. J Am Chem Soc 2013, 135, 10396-10403, doi:10.1021/ja4028346.
Namba et al. suggest, that repression of TET protein activity could play significantly part in EBV infection.
Namba-Fukuyo H, Funata S, Matsusaka K, Fukuyo M, Rahmutulla B, Mano Y, Fukayama M, Aburatani H, Kaneda A. TET2 functions as a resistance factor against DNA methylation acquisition during Epstein-Barr virus infection. Oncotarget. 2016 Dec 6;7(49):81512-81526. doi: 10.18632/oncotarget.13130. PMID: 27829228; PMCID: PMC5348409.
Author Response
Reviewer 3:
Open Review
Comments and Suggestions for Authors:
The presented topic is interesting and suitable for cancer. Nevertheless, in my opinion authors they, conceive the problem too narrowly. Most probably, Deferoxamine and Deferasirox can have significantly wider range of effect relevant to gene expression. Jumonji domain histone demethylases and TET proteins are α-ketoglutarate and Fe(II)-dependent dioxygenases. Decrease Fe(II) level by chelator can increase DNA and histone methylation. In the discussed hypoxia condition HIF-1a and TET proteins collaborate gene expression.
Badal, S.; Her, Y.F.; Maher, L.J., 3rd. Nonantibiotic Effects of Fluoroquinolones in Mammalian Cells. J Biol Chem 2015, 290, 22287-22297, doi:10.1074/jbc.M115.671222.
Palei, S.; Weisner, J.; Vogt, M.; Gontla, R.; Buchmuller, B.; Ehrt, C.; Grabe, T.; Kleinbölting, S.; Müller, M.; Clever, G.H.; et al. A high-throughput effector screen identifies a novel small molecule scaffold for inhibition of ten-eleven translocation dioxygenase 2. RSC Med Chem 2022, 13, 1540-1548, doi:10.1039/d2md00186a.
Chen, H.F.; Wu, K.J. Epigenetics, TET proteins, and hypoxia in epithelial-mesenchymal transition and tumorigenesis. Biomedicine (Taipei) 2016, 6, 1, doi:10.7603/s40681-016-0001-9.
Thank you for taking the time to carefully review our manuscript and raising some plausible alternative explanations for our findings and citing some of the relevant literature. We agree that DFO and DFX can affect the activities of the cellular TET and JMJD demethylases as well as the HIF-αs. We now discuss how these additional effects of these drugs may have influenced our findings (lines 509-530 of revised text).
In presence of ascorbate, iron deficiency can not be limited factor for the activity of TET proteins and probably for other α-ketoglutarate and Fe(II)-dependent dioxygenases.
Dickson, K.M.; Gustafson, C.B.; Young, J.I.; Züchner, S.; Wang, G. Ascorbate-induced generation of 5-hydroxymethylcytosine is unaffected by varying levels of iron and 2-oxoglutarate. Biochem Biophys Res Commun 2013, 439, 522-527, doi:10.1016/j.bbrc.2013.09.010.
Yin, R.; Mao, S.Q.; Zhao, B.; Chong, Z.; Yang, Y.; Zhao, C.; Zhang, D.; Huang, H.; Gao, J.; Li, Z.; et al. Ascorbic acid enhances Tet-mediated 5-methylcytosine oxidation and promotes DNA demethylation in mammals. J Am Chem Soc 2013, 135, 10396-10403, doi:10.1021/ja4028346.
We agree there is literature indicating that the ascorbate level can affect the iron requirement for the activities of some members of this class of enzymes. Thus, we performed some preliminary in vitro experiments to examine whether DFO-induced reactivation of EBV might be affected by the level of ascorbate in the medium. We found similar levels of EBV reactivation in SNU-719 cells regardless of whether ascorbate was present or not in the medium (data not shown). Regardless, we agree that the presence of vitamin C in the mice may have attenuated the effect of DFO- and DFX-induced EBV reactivation in the tumors. We now mention this possibility in the Discussion (lines 560-563).
Namba et al. suggest, that repression of TET protein activity could play significantly part in EBV infection.
Namba-Fukuyo H, Funata S, Matsusaka K, Fukuyo M, Rahmutulla B, Mano Y, Fukayama M, Aburatani H, Kaneda A. TET2 functions as a resistance factor against DNA methylation acquisition during Epstein-Barr virus infection. Oncotarget. 2016 Dec 6;7(49):81512-81526. doi: 10.18632/oncotarget.13130. PMID: 27829228; PMCID: PMC5348409.
We agree and have now included in the Discussion the role of TETs in EBV infection and reactivation (lines 522-530).
Given our responses above, we hope you now find our revised manuscript acceptable for publication in Cancers.
Round 2
Reviewer 1 Report
The authors have addressed the comments in a satisfactory manner.
Reviewer 3 Report
I have no serious objection, only cheking double spaces e.g., 548, 550, 552.